# A *Chlamydia psittaci* Outbreak in Psittacine Birds in Sardinia, Italy

**DOI:** 10.3390/ijerph192114204

**Published:** 2022-10-30

**Authors:** Gaia Muroni, Luigia Pinna, Elisa Serra, Valentina Chisu, Daniela Mandas, Annamaria Coccollone, Manuel Liciardi, Giovanna Masala

**Affiliations:** 1Zoonotic Pathology and OIEReference Laboratory for Echinococcosis, National Reference Center for Echinococcosis (CeNRE), Istituto Zooprofilattico Sperimentale della Sardegna, Via Duca degli Abruzzi n. 8, 07100 Sassari, Italy; 2S.C. Complex Territorial Diagnostic Structure of Cagliari, Istituto Zooprofilattico Sperimentale della Sardegna, 09067 Cagliari, Italy

**Keywords:** *Chlamydia psittaci*, *Chlamydiales*, *Avian psittacosis*, zoonoses, One Health, Italy

## Abstract

*Chlamydia psittaci* is an intracellular bacterium belonging to the *Chlamydiaceae* family. It is the ethiologic agent of psittacosis, an occupational zoonotic disease that mainly concerns people who work in close contact with birds that represent the main infection route for human transmission. In Italy, information about this disease is lacking. This study is the first case of avian chlamydiosis reported from a pet shop in Sardinia, Italy. *Chlamydia psittaci* detected in psittacine birds by molecular analysis, direct immunofluorescence test together with anatomo-pathological observed lesions, highlighted the importance of focusing the attention over this underestimated zoonosis in a “One Health” prospective.

## 1. Introduction

*Chlamydiae* are obligate intracellular bacteria with a worldwide distribution causing a wide range of diseases in human hosts, livestock, and companion animals as well as wildlife and exotic species [1,2]. Chlamydiosis is considered an occupational zoonosis of pet shop owners, zookeepers, veterinarians, abattoir workers, exotic and domestic bird breeders, pet bird enthusiasts [2]. Chlamydiosis in animals can range from asymptomatic infection to severe diseases with life-threatening illness depending on the host species affected and chlamydial species involved [3]. The family of *Chlamydiaceae* contains a single genus, *Chlamydia* including 11 species: *C. abortus*, *C. avium*, *C. caviae*, *C. felis*, *C. gallinacea*, *C. muridarum*, *C. pecorum*, *C. pneumonia*, *C. psittaci*, *C. suis*, *C. trachomatis* [4]. The two well-known species, *C. abortus* and *C. psittaci* are important zoonotic pathogens of livestock and avian species, respectively [5].

*Chlamydia psittaci* has at least 8 serotypes and 9 genotypes (A—F, E/B, M56 and WC), they have been identified by ompA marker analysis. These genotypes are specific for different hosts: generally, A and B genotypes infect Psittaciformes birds and pigeons, respectively, C genotype was isolated from ducks and geese, D genotype was isolated from turkeys; in turkeys and Psittacidae birds were found also F genotype. E Genotype has several hosts: it has been isolated from pigeons, ratites, ducks, and turkeys. Finally, E/B, WC and M56 genotypes were detected in ducks, Wolfsen cattle and muskrats, respectively. All genotypes are potentially transmissible to humans, and they can cause severe disease and even death [6,7,8,9].

The transmission of the infectious agent among birds occurs by direct contact, through the release of nasal excretions or contact with contaminated faeces and concerns the most susceptible birds [10]. In birds, the disease can manifest itself in acute, subacute, and chronic forms with symptoms including anorexia, diarrhea, lethargy, weight loss, and sometimes it presents only mucopurulent or serious oculonasal discharge. In severe cases, dark green faeces, anorexia, dehydration, dyspnea, and death [2].

Human psittacosis is a relatively rare zoonosis, with a clinical course that is usually severe [11]. The disease can be asymptomatic or symptomatic with flu-like syndrome (such as headache, fever, chills, malaise and myalgia, non-productive cough and breathing difficulties) up to severe atypical pneumonia [12,13]; cases of fulminant psittacosis, with multi-organ failure, are more rarely reported. It is rarely a disease with a fatal outcome, if treated with adequate antibiotics.

Even though the characterization of *C. psittaci* on genotype level from isolated cultures represents a key subject to understand epidemiology and clinical impact of this bacterium in animals and eventually in humans, for a rapid and specific diagnosis is suggested to perform the polymerase chain reaction (PCR). In fact, the culture of *C. psittaci* represents a laborious, time-consuming technique, and it requires a level 3 biosafety facility in laboratory [14]. Serological tests are used for diagnosis, which are still non-specific and difficult to interpret [6,7,8]. Other methods, such as immunochemistry, are actually used for the detection of this pathogen in cytological and histological preparations [14].

Despite chlamydiosis is a relatively common disease in parrots worldwide, avian psittacosis has not yet been described in parrots in Italy. However, few epidemiological cases of the disease are reported in other birds in this country: Magnino et al. [15] studied avian chlamydiosis in pigeons, Donati et al. [16] analysed collared doves’ cases and Di Francesco et al. [17] studied the pathology on corvids. Until now, there are no reports about psittacine birds that have been affected by *C. psittaci* infections in Sardinia. This work reports an outbreak of avian psittacosis in a pet shop and focuses the attention on the importance of reporting this type of zoonosis, the assessment of different samples and the use of different diagnostic tests for the identification of *C. psittaci*, which are currently underestimated.

## 2. Case Presentation

At the end of November 2021, a suspected outbreak of avian chlamydiosis occurred in psittacine birds housed in a pet shop located in South Sardinia, near Cagliari. The population of psittacine birds was represented by the following genera: *Pionites*, *Psittacus*, *Psittacula* and *Agapornis*, as illustrated in Table 1.

Each bird was lodged in contiguous cages within the same pet shop. Two birds died during this outbreak. The first one was an adult (three-years-old female) of Black-headed Caique Parrot (*Pionites melanocephalus*), whose carcass was sent to the laboratory of the Istituto Zooprofilattico Sperimentale (IZS) della Sardegna, section of Cagliari for post-mortem examination. The pet shop owner reported that the death of the bird was unexpected since the bird did not show evident clinical signs, except for the isolation of the animal from the other parrots that were in the cage. Ten days after the Caique, a rosy-faced lovebird (*Agapornis roseicollis*) died, and its carcass was sent to IZS della Sardegna for further analysis. The newly introduced parrot was kept alone in a separate cage, and it was quarantined for an undefined period. The medical history reported anorexia, lethargy, and dyspnea. The owner admitted that when the bird did not completely eat the food, the leftovers were distributed in other cages. Unfortunately, the necropsy of the *Agapornis roseicollis* was not possible due to autolysis of internal organs. On the other hand, the anatomo-pathological analysis was only performed on the Caique and showed a clear presence of regurgitation in the oral cavity, hyperemic and congestive neck region (Figure 1) associated with ectasia of blood vessels.

The coelomic cavity of the Caique was opened and examined. Kidneys and lungs were congested, and pulmonary edema was also present (Figure 2). Slightly enlarged and congestive liver was observed with hemorrhagic petechiae on the surface.

All necroscopy procedures and sample collection followed institutional guidelines and regulations. Microscopic analysis, performed on the intestinal mucosa and proventricolus, did not show any suggestive evidence for chlamydiosis. The intestine tissues were evaluated by Giemsa method, and it was reported as negative. Portions of liver were aseptically collected and processed by direct immunofluorescence test (IFD; IMAGEN Chlamydia test; Oxoid Ltd., Basinstoke, UK) for the determination of *Chlamydia* spp., and they gave positive signal as shown in Figure 3.

According to the established positivity for *Chlamydia* spp., the disease was notified by the veterinarian to the Regional Public Health Unit that ordered the temporary closure of the pet shop. The disinfection treatment of the room was performed by Neo Steramina G (Formevet, Italia). The remaining birds were moved to different cages and cloacal swabs were taken from each of them before antibiotic treatment and sent to the laboratories of the IZS della Sardegna, Sassari section, for IFD test and molecular analysis. The immunofluorescence test from cloacal swab samples collected from the living birds was negative.

Moreover, DNA was extracted from samples listed in Table 1 using the DNeasy Blood and Tissue Kit (QIAGEN, Hilden, Germany) following the manufacturer’s instructions.

Extracted DNA was stored at 4 °C until used in PCR amplification assays. At first, the DNA was subjected to conventional PCR by using 16SFor2 (5′ CGTGGATGAGGCATGCAAGTCGA 3′) and 16SRev6 (5′ ATCTCTCAATCCGCCTAGACGTCA 3′) primers which amplify a 256-bp fragment of the 16S rRNA gene (Centro di Referenza Nazionale Clamidiosi, IZS della Lombardia e dell’Emilia Romagna, Sezione Diagnostica di Pavia, Italy). DNA of Chlamydia abortus isolated from aborted fetus in Sardinia [18] was used as a positive control. Also, DNA extractions from a Chlamydia negative sample were used in each experiment as a negative control. PCR amplifications were performed in an automated DNA thermal cycle (GeneAmp PCR Systems 2400 and 9700; Applied Biosystems, Courtaboeuf, France) with an initial denaturation at 95 °C for 15 min, followed by 30 cycles of denaturation at 94 °C (60 s), annealing at 60 °C (30 s), and extension at 72 °C (60 s), followed by a final extension at 72 °C for 5 min. PCR products were then separated by electrophoresis in 1.5% agarose gel stained with SYBR Safe DNA Gel Stain (Invitrogen) and examined over UV light in an ImageMaster VDS-CL Systemvisualised (Amersham Biosciences Europe GmbH, Milano, Italy). Amplicons were purified using the QIAquick PCR Purification Kit (Qiagen, Hilden, Germany) according to the manufacturer’s protocol, and bidirectionally sequenced by DNA sequencing kit (dRhodamine Terminator cycle sequencing ready reaction; Applied Biosystems), according to the manufacturer’s instructions. Chromatograms were edited with Chromas 2.2 (Technelysium, Helensvale, Australia), and aligned with CLUSTALX [19] to assign sequences to unique sequence types. Sequences obtained were then compared with those of the Chlamydiales order present in the GenBank database using the BLAST search tool. Five out of six cloacal swabs examined in this study contained chlamydial DNA as represented in Table 1.

Among them, 16S rRNA Chlamydiales genotypes identified from one white-bellied parrot and one rose-ringed parakeet shared 100% sequence identity with *C. psittaci* strains isolated from Australian ungulates (accession number MK112573). The sequence obtained from the liver sample collected from the black-headed parrot was 100% identical to sequences of *C. psittaci* (Accession Number: MK112573; MK112572) which was identified as the closest match by BLASTn. Unfortunately, four templates did not show an excellent quality sequence and the identification of the bacterial species was not possible.

In order to confirm the presence of *C. psittaci*, 16S rRNA positive samples were additionally tested with oligonucleotide primers based on ompA gene specific for *C. psittaci* species [20]. Each reaction consisted of 12.5 µL of QuantiTect Probe PCR Master Mix (Qiagen, Toronto, Canada; 1× final concentration), Milli-Q water RNAse-free (9 µL), 2.5 µL each forward and reverse primer (2 µM) and 1 µL of DNA template, in a final volume of 25 μL. A negative control (DNA extracted from water) and a positive control (DNA of *C. psittaci* provided by the (Centro di Referenza Nazionale Clamidiosi, IZS della Lombardia e dell’Emilia Romagna, Sezione Diagnostica di Pavia, Italy) were included in each PCR test. The same conditions as described above were used for *C. psittaci*, changing only the number of cycles that were 40.

The presence of *C. psittaci* was confirmed in all tested samples. Based on the known susceptibility of *C. psittaci* to tetracyclines and according to a recent study, the treatment consisted in the administration of doxycycline at doses of 35 mg/kg q 24 h) for 21 days per os [21].

## 3. Discussion

This study documents the first detection of a chlamydiosis outbreak in psittacine birds in a pet shop in Sardinia, Italy. In this study, six out of seven (85.7%) of the psittacine birds were positive for genotypes based on outer membrane gene A (ompA) specific for *C. psittaci* strains. It was in line with previous studies in which ompA gene was used for *C. psittaci* identification [22]. The data from the present study showed a higher number of positive samples detected from cloacal swabs that are usually used for diagnosis of *C. psittaci* from live birds [2]. The nasal excretion, direct contact with plumage, dried feces and tissues of infected birds represent the most important route of human infection [23]. Since *C. psittaci* is highly infective, employers should implement security measures and adopt the most useful strategy for reducing the risk of transmission (e.g., the clear separation of water and food, which can represent a vehicle for the transmission of the etiological agent, or proper quarantine of diseased and newly introduced birds. Results obtained from this study confirm that PCR analysis had significantly better sensitivity than direct immunofluorescence test for the detection of chlamydial infection. Specifically, only the liver tissue from the Black-headed Caique Parrot (*Pionites melanocephalus*) tested positive after direct immunofluorescence test; whereas 6 samples resulted PCR-based Chlamydiales identification (Table 1). Although the choanal swabs are considered the most reliable in the early stage of the infection, our results demonstrated that cloacal swabs collected from the live birds, can be successfully used for detection of *C. psittaci* and it was in accordance with other studies in which the isolation of chlamydial agents from this matrix was used [3,24,25].

On necropsy in birds, the gross post-mortem lesions like splenomegaly, hepatomegaly, air sac changes, and fibrinous pericarditis will be noticed, but not pathognomonic [3]. Lesions are usually absent in latent infection. There are no ‘gold standard’ tests to identify recent infections. However, it has been known that the type of test performed could be influenced by the type of sample used. PCR method can offer a rapid and specific diagnosis; in addition, nucleic acid–based tests can provide capacity for strain genotyping.

The results obtained suggest that the introduction of psittacine birds of unknown origin represent a potential risk for animal and public health and this procedure is strongly discouraged. Imported birds should be subjected to a quarantine period and only healthy birds tested negative for *C. psittaci* by PCR and/or serological tests should be sold or imported [2]. Separating birds that test positive from those that tested negative is a recommended control measure [6]. Moreover, infected birds must be treated according to methods approved by the competent authority. Their quarantine must be extended by 2 months, at the end of this period, they must be retested. If positivity persists, quarantine is extended for another 2 months, and another treatment should be performed [14]. However, these procedures are not routinely followed by pet shop owners and positive cases among pets are frequent.

As referred by the owner of the pet shop, the death of the Caique occurred 2 weeks after the arrival of *Agapornis roseicollis* (less than one year old) to the pet shop given to the pet shop owner by a customer unable to take care of it. The details of the previous owner and the bird’s origin are unknown. Since chlamydiosis represents a high risk for the other birds and for human’s preventive measures are the key point in the control of psittacosis. In this study, the use of personal protective equipment (PPE), mandatory during the emergency of COVID-19, has been effective in protecting the employers of the pet shop from the *C. psittaci* infection and reducing the risk of contamination in the environment. As a consequence, no other cases of chlamydiosis were notified to the Regional Public Health Unit. The disease is rarely fatal in humans if associated with rapid diagnosis and adequate treatment: early diagnosis and the timely identification of appropriate antibiotic treatment reduce the rate of morbidity and mortality of chlamydiosis in humans [26,27]. However, although chlamydiosis is included among the notifiable diseases in the veterinary field and among the notifiable occupational diseases, this infection is not routinely sought and its prevalence is often underestimated [21,28]. Moreover, the identification of infected animals is challenging because animals can shed the bacterium without being symptomatic.

The eradication of the disease is extremely difficult in the absence of a vaccine. However, there are several prevention measures that can be adopted to mitigate the incidence of chlamydiosis; for example, the use of protective clothing and masks, correct disposal of infected material, quarantine of imported birds, adequate disinfection of the potentially contaminated areas because *C. psittaci* can persist for prolonged periods in the environment [29]. In some cases, pet shop owners include tetracycline in the drinking water of birds for sale (especially for psittacine birds such as parrots) to keep them healthy until sold [10]. However, this wrong practice could often generate resistant strains of bacteria that may become established in the psittacine birds [30]. Beyond these strategies for prevention, a more effective monitoring and reporting activities applying a One Health approach, as recommended at international level, would in any case be desirable.

## 4. Conclusions

The results of this study suggested baseline information to address future epidemiologic, pet shop management and public health policies for the prevention of psittacosis inside and outside Sardinia, Italy. There is a great need for a deeper investigation of unknown importation of pet birds potentially infected, to prevent the risk of human and pet birds’ infections. Unfortunately, illegal trade still occurs, so whenever incoming birds have a history of inadequate health management, adequate control measures and prophylaxis are essential. Specific training activities to the staff of pet shops organized by health authorities or veterinarians would be strategic to give instructions on the importance of biosecurity measures aimed at reducing the risk of introduction of *Chlamydia* spp., and other zoonotic agents. The disease is more frequent than previously thought. The efficacy of vaccines against *C. psittaci* and the evaluation of this pathogen shedding in birds should be the major subject of future research, even though investigations have been and are ongoing [2,31].

Currently, tests on nucleic acid amplification such as conventional or real time PCR are considered the state of art methods to diagnose chlamydial infections in animals and humans due to their sensitivity and specificity compared to other tests (e.g., serological analysis or immunofluorescence and immunohistochemical staining) [2].

Laboratories should include this pathogen in routine diagnosis and European standards for diagnosis of *C. psittaci* infections in birds and humans must be followed strictly to lead a common pattern.

## Figures and Tables

**Figure 1 ijerph-19-14204-f001:**
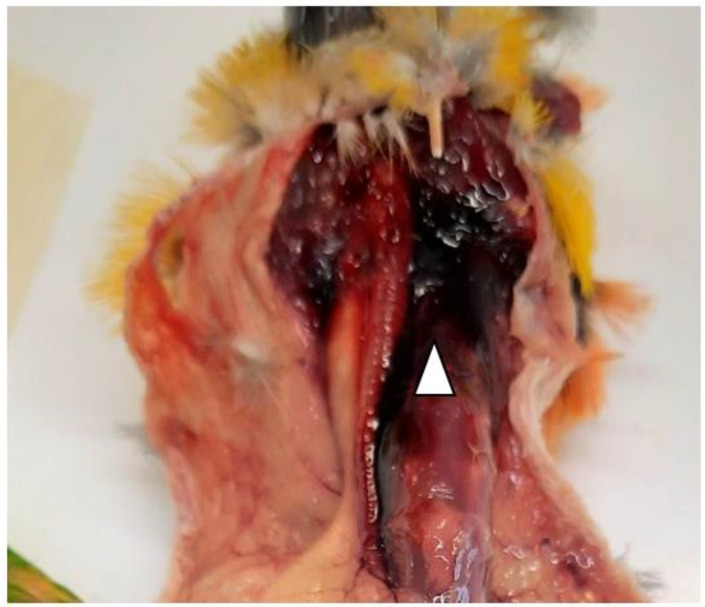
Hyperemic and congestive neck region (white arrowhead) identified during the necropsy of the Caique.

**Figure 2 ijerph-19-14204-f002:**
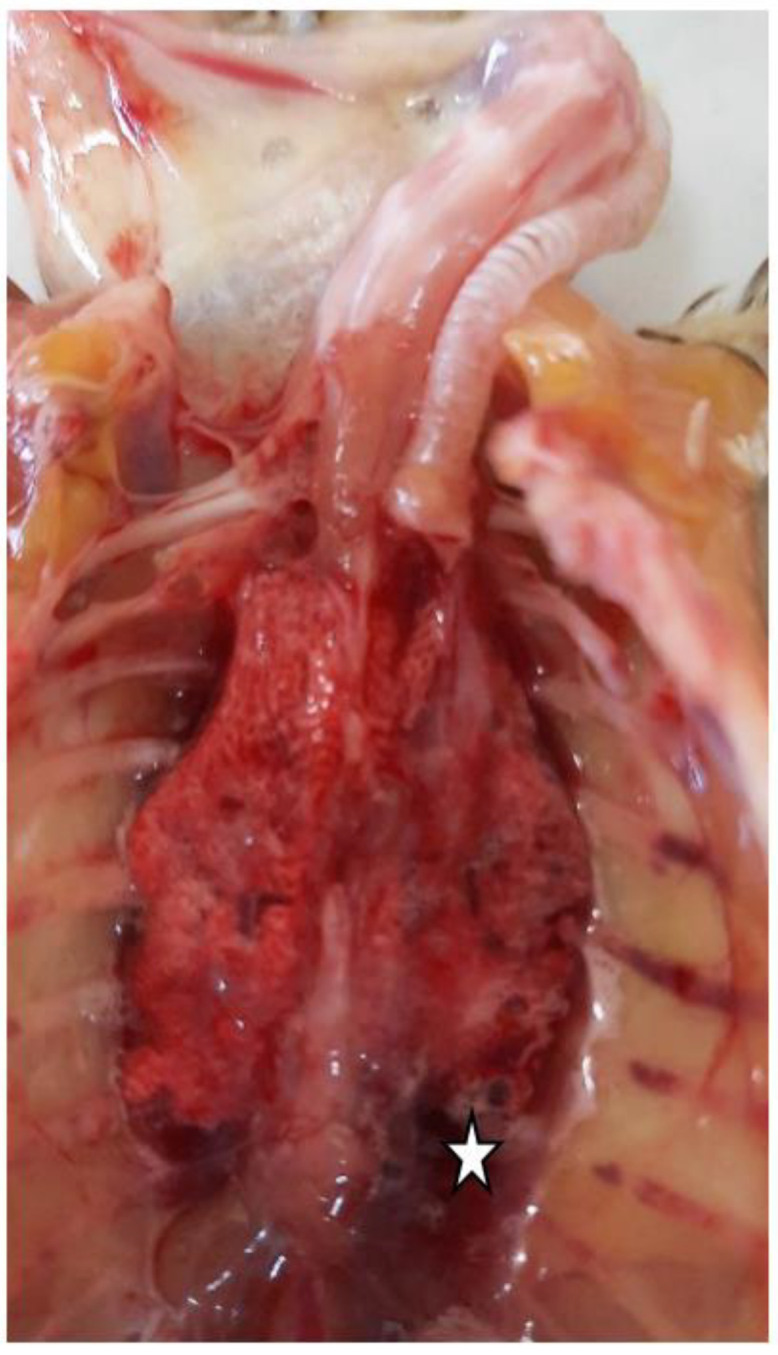
Congestive lungs and pulmonary edema (white star) identified during the necropsy of the Caique.

**Figure 3 ijerph-19-14204-f003:**
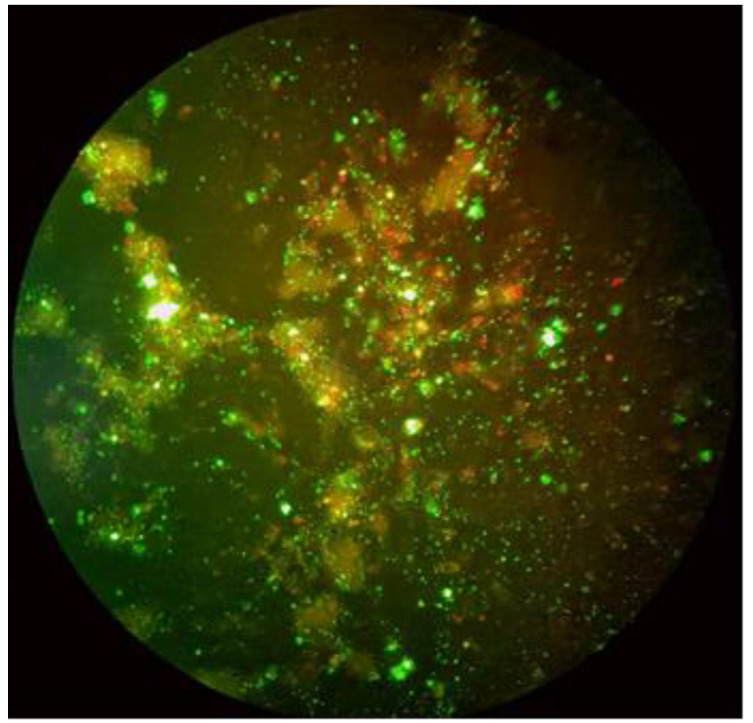
Liver cell smear positivity to *Chlamydia* spp. (apple green) detected by a direct Immunofluorescence test (40×).

**Table 1 ijerph-19-14204-t001:** Origin of samples investigated in this study, and summary of IFD, PCR (16S rRNA and ompA genes) and BLASTn maximum identities of the 16S rRNA gene sequences.

Host (Scientific Name)	Age	Death	Specimen Type	IFD	16S rRNA PCR	BLASTn Maximum Identities	OmpA PCR
Black-headed parrot (*Pionites melanocephalus*)	3	No	Cloacal swab	−	+	−	+
White-bellied parrot (*Pionites leucogaster*)	3	no	Cloacal swab	−	+	−	+
White-bellied parrot (*Pionites* *leucogaster*)	3	no	Cloacal swab	−	+	100% *Chlamydia psittaci*	+
Grey parrot (*Psittacus* *Erithacus*)	9	no	Cloacal swab	−	−	−	−
Rose-ringed Parakeet (*Psittacus krameri*)	9	no	Cloacal swab	−	+	100% *Chlamydia psittaci*	+
Rosy- faced lovebird (*Agapornis roseicollis*)	Unknown	yes	Cloacal swab *	−	+	−	+
Black-headed parrot *(Pionites melanocephalus)*	3	yes	Liver	+	+	100% *Chlamydia psittaci*	+

* Sample collected when the psittacine bird was still alive.

## Data Availability

Not applicable.

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
