# Peer review of "A Chlamydia psittaci Outbreak in Psittacine Birds in Sardinia, Italy"

_ijerph, 2022, doi:10.3390/ijerph192114204_

Round 1

Reviewer 1 Report

Chlamydiosis is a relatively common disease in parrots around the world. In fact, I was surprised that chlamydia has not yet been described in Italy, where bird breeding is well developed. This is likely due to legislation that severely impose on breeders or store owners the harsh consequences of officially disclosing and reporting the disease. Therefore, such cases are rarely reported and published.

Abstract

In the summary, there is little information about the case itself, it is worthwhile to complete this.

“Psittacosis” is a human disease caused by Chlamydia psittaci, when the bacterium causes disease in birds, this disease is called “avian chlamydiosis”.

Introduction

Chlamydiosis is a zoonosis, but it occurs relatively rarely, it is worth emphasizing in the introduction, confirming with quotations.

L23-30: No citations to the statements presented. Publication [1] does not contain them.

L30: C. gallinacean,- should be C. gallinacea

L31: C. pneumonia,- should be C. pneumonia

L32: There is no citation.

L27: Asyntomptomatic- should be “asymptomatic”

L36: Psittacidae- Since chlamydiosis is common in other species of psittacine birds like cockatiel (mentioned in research)  as well, it would be better to include the entire order of Psittaciformes than only Psittacidae family. The term psittacine birds may also be used.

L44: should be “faeces”

L46- 47: ..lethargy, and weight loss,..- should be “..lethargy, weight loss and sometimes only mucopurulent or serous oculonasal discharge.”

L48: “stools”- should be “faeces”

L48: “and death may occur” - in acute cases of chlamydiosis in parrots without treatment, death is almost certain. So it would be better to write "and death"

L48: The cited article [7] is not appropriate. It is about serology, not clinical symptoms.

L51: According to citation [8] should be: “atypical pneumonia” also in:  https://www.archbronconeumol.org/en-atypical-chlamydia-psittaci-pneumonia-four-articulo-S1579212917300897

Case presentation:

L80- sentence should be changed, e.g.: Two days after the caiqe, the rosy-faced lovebird died, and was also sent to IZS della Sardegna for further analysis.

L82- L84: The post-mortem lesions mentioned by you are not typical of chlamydia. The hematoma visible in Fig. 1. may be the result of a mechanical trauma, and the photo does not show any regurgitated content. Hasn't the already dying animal been culled with cervical vertebrae dislocation?

Fot. 2. The neck looks normal, pulmonary edema is not accompanied by severe congestion typical of pneumonia. These photos do not add anything, only in Fig. 2 kidneys that are swollen and pale.

L92: Birds do not have an abdominal cavity, only a coelomic cavity (no diaphragm). The kidneys (at least in the photo) are swollen and pale. Photographs of the liver and descriptions of the spleen, which is usually significantly enlarged in chlamydiosis, are very lacking.

L93: „hemorrhagic catarrhal enteritis” should be confirmed by histopathological examination if it cannot only be suspected that the condition has occurred. I would like to mention that the presence of blood in the intestines of small birds can sometimes be caused by starvation (sick birds may not eat food or have problems accessing it).

L94: “Slightly enlarged (hepatomegaly)”- you have to decide on something or write down which parrot had a slightly enlarged liver and which had hepatomegaly.

L100- 101: should be “proventriculus”, the authors emphasize at this point the examination of the gastrointestinal tract as if there were to be any pathognomonic changes? I would pay more attention to the parenchymal organs and the respiratory system. Was only the gastrointestinal tract examined?

L100- 102 . These sentences should be combined into one about the use of the microscopic examination method for chlamydia and its result.

L107 Chlamydia –italics

L111: Is peramycin G a trade name for a disinfectant? If so, please specify the manufacturer and the place of production.

L114: The immunofluorescence test from samples collected….

L125-138: Too detailed description of this PCR test. While the second PCR test is very modestly described.

L 115: I would replace "turned out" with "was".

L151-154 Incomprehensible sentence.

L159- 161. A very low dose of doxycycline was used. According to Carpenter Exotic Animals Formulary, the dose for parrots is 35-50 mg/kg body weight per 24h.

 Has the effectiveness been confirmed after the end of the therapy? Better than "by mouth" will be "orally" or per os.

Have shop employees been tested for chlamydia ?

Were only the birds sold there? Chlamydia psittaci can also be pathogenic to other animals.

Table 2. Only one table can be inserted instead of two tables. The species in Table 1 and Table 2 do not match. Cockatiel is only in table 2.
What does the word "Pool" mean in this context?

L122- 123: The phrase "abortion product" should be replaced with, for example, "aborted fetus" or " placenta" depending on the type of sample.

Disscussion

L163: “psittacosis”- should be “chlamydiosis”

L164- 165: This sentence should be corrected.

L167- 169: nasal or from choane

L168: A swab can be taken from the throat and choane (nose) at one time -nosopharyngeal swab. It can also be taken only from the throat or only from the choane. The authors include the former in the table, and the latter in the discussion. Please clarify.

L174: „or the isolation of new introduction of imported animals” this sentence should be corrected.

L176: "chlamydial" in lowercase

L176: Has direct immunofluorescence been used to examine other organs of dead parrots? If so, what and with what effect?

L179- 181: Table 1 shows that the origin of most of these birds was known. Knowing the origins of animals does not guarantee that they are not disease carriers.

L181- 183: Imported birds should be quarantined and only healthy birds tested negative for chlamydia by PCR and / or serological tests (Ref) should be sold.

(The whole problem, however, is that the price of a lovebird, for example, is lower than the price of a chlamydiosis test:)

L183- 184: “Symptomatic animals should be tested for psittacosis by PCR using pharyngeal, conjunctival, or cloacal swabs”- unnecessary sentence.

L186: Citation is necessary.

L191- 194: Most of this information should be included in the case presentation section.

L193- 194: Lovebirds are grain eaters, fruits are only a variety of diet. If the bird was not properly fed and additionally under social stress, it could lead to immunosuppression and disease development after infection from carrier birds, so there is no certainty as to the source of the infection.

L201- 206: This information is already in the introduction.

L218: Please find a citation for how long Chlamydia psittaci may persist in the environment.

L219- 220: It is not properly emphasized that this is a wrong action and why.

Conclusions

227-229: Writing about illegal importation and trade in the aspect of the presented case is quite an exaggeration. Someone gave the lovebird to the pet shop, because he did not want to have it longer, that's all …

234- 235: Research into chlamydia vaccines in birds has been and is ongoing. Relevant publications can be cited.

References:

L287 missing abbreviation of the journal.

Author Response

Author's response to Reviewer's Comments (Reviewer 1)

Chlamydiosis is a relatively common disease in parrots around the world. In fact, I was surprised that chlamydia has not yet been described in Italy, where bird breeding is well developed. This is likely due to legislation that severely impose on breeders or store owners the harsh consequences of officially disclosing and reporting the disease. Therefore, such cases are rarely reported and published.

Abstract

In the summary, there is little information about the case itself, it is worthwhile to complete this.

Response: More information about the case itself has been added in the summary

Psittacosis” is a human disease caused by Chlamydia psittaci, when the bacterium causes disease in birds, this disease is called “avian chlamydiosis”.

Response: Corrected

Introduction

Chlamydiosis is a zoonosis, but it occurs relatively rarely, it is worth emphasizing in the introduction, confirming with quotations.

L23-30: No citations to the statements presented. Publication [1] does not contain them.

Response: References have been added

L30: C. gallinacean,- should be C. Gallinacea

Response: Changed

L31: C. pneumonia,- should be C. pneumonia

Response: Changed

L32: There is no citation.

Response: It has been added

L27: Asyntomptomatic- should be “asymptomatic”

Response: Corrected

L36: Psittacidae- Since chlamydiosis is common in other species of psittacine birds like cockatiel (mentioned in research) as well, it would be better to include the entire order of Psittaciformes than only Psittacidae family. The term psittacine birds may also be used.

Response: Done

L44: should be “faeces”

Response: Corrected

L46- 47: ..lethargy, and weight loss,..- should be “..lethargy, weight loss and sometimes only mucopurulent or serous oculonasal discharge.”

Response: Thank you for your suggestion. The sentence has been modified

L48: “stools”- should be “faeces”

Response: Done

L48: “and death may occur” - in acute cases of chlamydiosis in parrots without treatment, death is almost certain. So it would be better to write "and death"

Response: Done

L48: The cited article [7] is not appropriate. It is about serology, not clinical symptoms.

Response: Changed

L51: According to citation [8] should be: “atypical pneumonia” also in: https://www.archbronconeumol.org/en-atypical-chlamydia-psittaci-pneumonia-four-articulo-S1579212917300897

Response: Added

Case presentation:

L80- sentence should be changed, e.g.: Two days after the caiqe, the rosy-faced lovebird died, and was also sent to IZS della Sardegna for further analysis.

Response: Changed

L82- L84: The post-mortem lesions mentioned by you are not typical of chlamydia. The hematoma visible in Fig. 1. may be the result of a mechanical trauma, and the photo does not show any regurgitated content. Hasn't the already dying animal been culled with cervical vertebrae dislocation?

Response: I agree with your observation. The lesions reported could be compatible with other avian diseases of bacterial and viral origin. These lesions were observed during necropsy, including hyperemia and congestion in the neck region; the medical history reports the spontaneous death of the animal and excludes trauma or killing by euthanasia. In fact, no fracture or dislocation of the cervical vertebrae was observed during the necropsy. The text has been modified and adapted to Fig. 1.

Fot. 2. The neck looks normal, pulmonary edema is not accompanied by severe congestion typical of pneumonia. These photos do not add anything, only in Fig. 2 kidneys that are swollen and pale.

Response: We agree with your comments and for this reason we preferred to delete this figure that does not add nothing.

L92: Birds do not have an abdominal cavity, only a coelomic cavity (no diaphragm). The kidneys (at least in the photo) are swollen and pale. Photographs of the liver and descriptions of the spleen, which is usually significantly enlarged in chlamydiosis, are very lacking.

Response: The text has been modified according to your suggestion. Unfortunately, we did not take a picture of the liver that was enlarged and showed petechial hemorrhages on the surface (data not shown).

L93: hemorrhagic catarrhal enteritis” should be confirmed by histopathological examination if it cannot only be suspected that the condition has occurred. I would like to mention that the presence of blood in the intestines of small birds can sometimes be caused by starvation (sick birds may not eat food or have problems accessing it).

Response: The text has been changed. No anomalies concerning nutrition were observed or reported even in sick animals.

L94: “Slightly enlarged (hepatomegaly)”- you have to decide on something or write down which parrot had a slightly enlarged liver and which had hepatomegaly.

R. The text was corrected by eliminating hepatomegaly.

L100- 101: should be “proventriculus”, the authors emphasize at this point the examination of the gastrointestinal tract as if there were to be any pathognomonic changes? I would pay more attention to the parenchymal organs and the respiratory system. Was only the gastrointestinal tract examined?

Response: The text has been corrected. Only portions of the intestine and liver were examined, it was not possible to carry out the examination of other parenchymal organs due to autolysis.

L100- 102. These sentences should be combined into one about the use of the microscopic examination method for chlamydia and its result.

Response: Thank you for your suggestion but we rather keep the sentence as it was originally.

L107 Chlamydia –italics

Response: It has been changed.

L111: Is peramycin G a trade name for a disinfectant? If so, please specify the manufacturer and the place of production.

R. The text has been modified with the correct name: Neo Steramina G (Formevet Italy).

L114: The immunofluorescence test from samples collected…

Response: The text was modified.

L125-138: Too detailed description of this PCR test. While the second PCR test is very modestly described.

Response: More information on the second PCR has been added.

L 115: I would replace "turned out" with "was".

Response: The author replaced “turned out” with “was”

L151-154 Incomprehensible sentence.

Response: The text was modified and now reads:

Among them, 16S rRNA Chlamydiales genotypes identified from one white-bellied parrot and one Rose-ringed Parakeet shared 100% sequence identity with C. psittaci strains isolated from Australian ungulates (accession number MK112573). The sequence obtained from the liver sample collected from the Black-headed parrot was 100% identical to sequences of C. psittaci (Accession Number: MK112573; MK112572) which was identified as the closest match by BLASTn.”

L159- 161. A very low dose of doxycycline was used. According to Carpenter Exotic Animals Formulary, the dose for parrots is 35-50 mg/kg body weight per 24h.

Response: Corrected, it was a type.

Has the effectiveness been confirmed after the end of the therapy? Better than "by mouth" will be "orally" or per os.

Response: After the therapy, no additional exams were performed on animals due to the refusing of the owner of the petshop.. “By mouth” was corrected with per os.

Have shop employees been tested for chlamydia ?

Response: There are no reports of human chlamydiosis in the shop. The surveillance of the staff in the shop is under responsibility of the health authority of human medicine (to which avian chlamydiosis has been reported). However, to our knowledge, no checks have been carried out.

Were only the birds sold there? Chlamydia psittaci can also be pathogenic to other animals.

Response: As reported by the pet shop owner, only birds were in the pet shop.

Table 2. Only one table can be inserted instead of two tables. The species in Table 1 and Table 2 do not match. Cockatiel is only in table 2. What does the word "Pool" mean in this context?

Response: Table one and two have been now merged and all missing or unclear information has been modified/added in the new Table 1.

L122- 123: The phrase "abortion product" should be replaced with, for example, "aborted fetus" or " placenta" depending on the type of sample.

Response: “abortion product” has been replaced.

Discussion

L163: “psittacosis”- should be “chlamydiosis”

Response: The author substituted “psittacosis” with “chlamydiosis”.

L164- 165: This sentence should be corrected.

Response: corrected

L167- 169: nasal or from choane

Response: corrected

L168: A swab can be taken from the throat and choane (nose) at one time -nosopharyngeal swab. It can also be taken only from the throat or only from the choane. The authors include the former in the table, and the latter in the discussion. Please clarify.

Response: The table and the text have been modified and in this new version

L174: „or the isolation of new introduction of imported animals” this sentence should be corrected.

Response: corrected

L176: "chlamydial" in lowercase

Response: Chlamydial has been written in lowercase.

L176: Has direct immunofluorescence been used to examine other organs of dead parrots? If so, what and with what effect?

Response: Everything has been clarified as shown in Table 1.

L179- 181: Table 1 shows that the origin of most of these birds was known. Knowing the origins of animals does not guarantee that they are not disease carriers.

Response: We agree with you and according with your comments and those of the Rev 2 we preferred to modify the table by merging Table 1 and Table 2 and by deleting the origin of parrots.

L181- 183: Imported birds should be quarantined and only healthy birds tested negative for chlamydia by PCR and / or serological tests (Ref) should be sold.

(The whole problem, however, is that the price of a lovebird, for example, is lower than the price of a chlamydiosis test:)

Response: according to your suggestion, the sentence has been modified and the reference was added.

L183- 184: “Symptomatic animals should be tested for psittacosis by PCR using pharyngeal, conjunctival, or cloacal swabs”- unnecessary sentence.

Response: The sentence has been deleted

L186: Citation is necessary.

Response: Thank you for your suggestion, citation is already present at the end of the period and the authors would like to keep it as it is.

L191- 194: Most of this information should be included in the case presentation section.

Response: Done

L193- 194: Lovebirds are grain eaters, fruits are only a variety of diet. If the bird was not properly fed and additionally under social stress, it could lead to immunosuppression and disease development after infection from carrier birds, so there is no certainty as to the source of the infection.

Response: The sentence has been modified.

L201- 206: This information is already in the introduction.

Response: We preferred to delete these information

L218: Please find a citation for how long Chlamydia psittaci may persist in the environment.

Response: Added

L219- 220: It is not properly emphasized that this is a wrong action and why.

Response: a new sentence has been added as follows:

However, this wrong practice could often generates resistant strains of bacteria that may become established in the psittacine birds

Conclusions

227-229: Writing about illegal importation and trade in the aspect of the presented case is quite an exaggeration. Someone gave the lovebird to the pet shop, because he did not want to have it longer, that's all …

Response: Sentence modified.

234- 235: Research into chlamydia vaccines in birds has been and is ongoing. Relevant publications can be cited.

Response: Citation added.

References:

L287 missing abbreviation of the journal.

Response: Added

Reviewer 2 Report

A brief summary:

The article entitled A Chlamydia psittaci Outbreak in Psittacine Birds in Sardinia, Italy” by Muroni et al. aims to draw attention to the need to report the occurrence of zoonotic diseases affecting commercial poultry that require sanitary control. This article is based on a C. psittaci outbreak diagnosed in Psittacine birds for the first time in Sardinia (Italy).

The main contribution of this study is to describe for the first time a zoonotic disease in an Italian region which reinforces the need for quarantine of birds to be sold as pets, with emphasis on birds of unknown origin. The study also assesses different samples and tests to identify C. psittaci in birds.

Broad comments:

Strengths of this study

·         The results provide interesting information on C. psittaci infection in Psittacine birds.

·         The manuscript is well structured.

·         Different analyses (anatomopathological, direct immunofluorescence and molecular) are performed for the identification and confirmation of the presence of the bacteria.

·         Bibliographic references are adequate for a case report, and they include 22 articles published between 2000 and 2022.

 Weaknesses of this study

 ·         The aims of the study should be improved. Although it is not an experimental study, the authors should include as other aims the evaluation of different samples and tests for the identification of C. psittaci.

·         The gross description of Caique needs to be improved. Besides, there is no mention of the lesions observed on the carcass of the second dead animal (Agapornis roseicollis).

·         The histopathological description is poor, and the authors do not indicate whether microscopic analysis was carried out on the two dead birds.

·  Discussion section should be improved by avoiding duplication of information included in the introduction.

·           Conclusion section needs to be improved. Conclusions related to the samples and tests for identification of the bacteria should be included.

·         See “Specific comments” below.

This article is an interesting study, but changes must be made for the final version.

Specific comments:

Some modifications or clarifications need to be made:

In general, ·   “ºC” must leave a space between number and symbol throughout the manuscript. e.g., 95 ºC (line 131) ·   “%” must leave a space between number and symbol throughout the manuscript. e.g., 1.5 % (page 137)

Introduction section

Line 24. Change "ad" to "and".

Line 33. Information is repeated: remove the sentence "belongs to the Chlamydiaceae family " because this information is already on line 29.

Line 60. The authors should include information on other methods for identification of chlamydia such as immunohistochemistry.

Line 67. The authors should add as other aims the assessment of different samples and tests for the identification of C. psittaci. This would improve the quality of the work.

 “Case presentation” section

Lines 80-81. Did the second dead bird show symptoms?

Line 83. Change “congested neck” to “congestive neck”. This modification must be made throughout the work.

Line 93. A gross diagnosis of catarrhal haemorrhagic enteritis is cited, but was it was confirmed on microscopic examination?

No gross lesions are mentioned on the second dead bird. This should be made clear.

Lines 100-103.

·         Was the microscopic analysis performed only on the intestine and proventriculus?

·         Other organs may show histopathological lesions associated with infection with this bacterium (aerosacculitis, pericarditis, pneumonia, hepatitis, splenitis, enteritis, conjunctivitis, encephalitis, nephritis, etc.). The reference only to the intestine and proventriculus is not enough. The authors should add more information on the histopathological findings, and they should make clear the microscopic findings in both carcasses.

·         Were the microscopic, Giemsa and IFD analyses performed on only one or both dead animals?

Line 105. Does the expression "they gave positive" indicates a result to both organs (liver and intestine) or to both dead animals?

Line 109. Chlamydia spp.

Line 114. Modify the following sentence to make it clearer because it is incomplete: The immunofluorescence test performed on samples collected from the living birds …..

Line 112. The authors should make it clear that the cloacal swabs were collected prior to treatment administration.

Line 121. Write well 4 ºC

Line 126-127. The authors indicate that the positive control is C. abortus DNA and refer to the literature reference [13]. This article is about Babesia and Theileira. Is this right?

Line 131-132. Write well 95 ºC and 72 ºC

Line 149. Chlamydia sp. or spp.?

Line 153. The sentence is incomplete: “…parrot, that lost their life during the ¿?..were 100 %”

Figure 1

·         The authors should point out where the regurgitated material is in the oral cavity. The picture is not clear in this respect and regurgitation cannot be seen (regurgitation happens when a mixture of gastric juices, and sometimes undigested food, rises back up the esophagus and into the mouth). Is it possible to change this photo for a clearer one?

·         Figure legend: the expression " noticeable ectasia of blood vessels" should be removed because such ectasia or dilatation is not observed macroscopically.

Figure 2. The picture also shows congestion in the lungs.

Figure 3. The “white star” does not show the edema clearly. Pulmonary edema in birds (unlike mammals) is identified macroscopically as a fluid clear or slightly blood-tinged visible which is located free in the thoracic cavity and on the cut surfaces of the lung. This vascular alteration should be better pointed out.

Figure 4.

·         Figure legend: Chlamydia in italics.

·         The authors should add more information on the color of the positivity and the magnification at which the photo was taken.

Table 1. The age of the animals could be added in this table to have more information on all birds.

Table 2.

·         The table heading should be modified: e.g, “……..summary of PCR (16S rRNA and omp A genes) and ……….”.

·         In row 8 of the table 2,  Cockatiel's specimen is incomplete.

·         The headings in the fourth and sixth columns should be changed to "16S rRNA PCR" and "omp A PCR", respectively, for a better understanding.

·         What is “Pool”?

·         Grey parrot was negative in all tests performed. Could the authors explain this fact?

Discussion section

Line 176. Change “Chlamydial infection” to “chlamydial infection”

Lines 176-179. The authors explained that:

Specifically, only the intestine and liver tissues from the Black-headed Caique Parrot (Pionites melanocephalus) tested positive after direct immunofluorescence test whereas 7 samples resulted PCR-based Chlamydiales identification (Table 2)”.

The authors should explain that these differences in positivity, in addition to the type of test performed, may also be influenced by the type of sample used (liver/intestine versus cloacal swab). Besides, they should compare their results with data from other studies.

Line 197. The authors should add the meaning of the abbreviation “PPE”.

Lines 201-206. Remove this information which is already in the “Introduction section”. This information is repeated unnecessarily.

Conclusions section

Other conclusions based on the results of this study should be included in this section of the paper. For example, highlighting the samples (cloacal and nasopharyngeal swab) and tests (16S rRNA and ompA PCR) that were best for identification of the infectious agent in this outbreak.

Line 233. Chlamydia spp.

References section

References 1 and 15. Should the name of the journal be written in abbreviations?

Author Response

Author's response to Reviewer's Comments (Reviewer 2)

The aims of the study should be improved. Although it is not an experimental study, the authors should include as other aims the evaluation of different samples and tests for the identification of C. psittaci.

Response: Thank you for your comments that allowed us to improve our manuscript. The aims of the study have been improved.

The gross description of Caique needs to be improved. Besides, there is no mention of the lesions observed on the carcass of the second dead animal (Agapornis roseicollis).

Response: we added more information on Caique gross description. Unfortunately, we have no information on the Agapornis whose internal organs were autolytic

The histopathological description is poor, and the authors do not indicate whether microscopic analysis was carried out on the two dead birds.

Response: histopathological description is now more complete and from the case report section is more clear that the second dead animal was not evaluated.

Discussion section should be improved by avoiding duplication of information included in the introduction.

Response: Improved

Conclusion section needs to be improved. Conclusions related to the samples and tests for identification of the bacteria should be included.

Response: according to your suggestion the conclusion section is now more complete.

Specific comments:

Some modifications or clarifications need to be made:

In general,·ºC” must leave a space between number and symbol throughout the manuscript. e.g., 95 ºC (line131)·%” must leave a space between number and symbol throughout the manuscript. e.g., 1.5 % (page137)

Response: Done

Introduction section

Line 24. Change "ad" to "and".

Response: “ad” was replaced with “and”.

Line 33. Information is repeated: remove the sentence "belongs to the Chlamydiaceae family " because this information is already on line 29.

Response: Sentence removed.

Line 60.The authors should include information on other methods for identification of chlamydia such as immunohistochemistry.

Response: Information added

Line 67. The authors should add as other aims the assessment of different samples and tests for the identification of C. psittaci. This would improve the quality of the work.

Response: Thanks for you suggestion. These aims have been added

“Case presentation” section

Lines 80-81. Did the second dead bird show symptoms?

Response: Medical history added.

Line 83. Change “congested neck” to “congestive neck”. This modification must be made throughout the work.

Response: changed.

Line 93. A gross diagnosis of catarrhal haemorrhagic enteritis is cited, but was it was confirmed on microscopic examination?

Response: corrected. In the final version of the manuscript and, according to the revisions suggested by the rev 1, this sentence has been modified.

No gross lesions are mentioned on the second dead bird. This should be made clear.

Response: It was not possible to carry out the necropsy examination of the Agapornis roseicollis specimen. This information has been specified in the text.

Lines 100-103.

Was the microscopic analysis performed only on the intestine and proventriculus?

Other organs may show histopathological lesions associated with infection with this bacterium (aerosacculitis, pericarditis, pneumonia, hepatitis, splenitis, enteritis, conjunctivitis, encephalitis, nephritis, etc.). The reference only to the intestine and proventriculus is not enough. The authors should add more information on the histopathological findings, and they should make clear the microscopic findings in both carcasses.

Were the microscopic, Giemsa and IFD analyses performed on only one or both dead animals?

Response: The information has been clarified, it was not possible to carry out microscopic analysis of other organs due to autolysis. IFD analyses results are shown in table 1.

Line 105. Does the expression "they gave positive" indicates a result to both organs (liver and intestine) or to both dead animals?

Response: sentence modified.

Line 109. Chlamydia spp.

Response:“spp.” was added

Line 114. Modify the following sentence to make it clearer because it is incomplete: The immunofluorescence test performed on samples collected from the living birds.

Response: The sentence has been modified.

Line 112. The authors should make it clear that the cloacal swabs were collected prior to treatment administration.

Response: information added.

Line 121. Write well 4 ºC.

Response: The authors wrote 4 ºC.

Line 126-127. The authors indicate that the positive control is C. abortus DNA and refer to the literature reference [13]. This article is about Babesia and Theileira. Is this right?

Response: The reference has been substituted.

Line 131-132. Write well 95 ºC and 72 ºC.

Response: the authors wrote well 95 ºC and 72 ºC

Line 149. Chlamydia sp. or spp.?

Response: sentence modified.

Line 153. The sentence is incomplete: “…parrot, that lost their life during the ¿?..were 100 %”

Response: the authors completed the sentence.

Figure 1.

The authors should point out where the regurgitated material is in the oral cavity. The picture is not clear in this respect and regurgitation cannot be seen (regurgitation happens when a mixture of gastric juices, and sometimes undigested food, rises back up the esophagus and into the mouth). Is it possible to change this photo for a clearer one?

· Figure legend: the expression " noticeable ectasia of blood vessels" should be removed because such ectasia or dilatation is not observed macroscopically.

Response: The text and caption of Figure 1 have been modified.

Figure 2. The picture also shows congestion in the lungs.

Response: Fig. 2 has been deleted because it is unclear.

Figure 3. The “white star” does not show the edema clearly. Pulmonary edema in birds (unlike mammals) is identified macroscopically as a fluid clear or slightly blood-tinged visible which is located free in the thoracic cavity and on the cut surfaces of the lung. This vascular alteration should be better pointed out.

Response: The lung parenchyma showed an abundant pinkish fluid leaking out when it was cut

Figure 4.

· Figure legend: Chlamydia in italics.

· The authors should add more information on the color of the positivity and the magnification at which the photo was taken.

Response: Corrections have been made.

Table 1. The age of the animals could be added in this table to have more information on all birds.

Response: added

Table 2.

The table heading should be modified: e.g, “……..summary of PCR (16S rRNA and omp A genes) and ……….”.

Response: added

In row 8 of the table 2,  Cockatiel's specimen is incomplete.

Response: completed

The headings in the fourth and sixth columns should be changed to "16S rRNA PCR" and "omp A PCR", respectively, for a better understanding.

Response:modified

What is “Pool”?

Response: in the new table 1 the organs and cloacal swabs obtained from birds are now modified.

Grey parrot was negative in all tests performed. Could the authors explain this fact?

Response: The discussion presents now more information

Discussion section

Line 176. Change “Chlamydial infection” to “chlamydial infection”.

Response:The change has been made

Lines 176-179. The authors explained that:

“Specifically, only the intestine and liver tissues from the Black-headed Caique Parrot (Pionites melanocephalus) tested positive after direct immunofluorescence test whereas 7 samples resulted PCR-based Chlamydiales identification (Table 2)”.

The authors should explain that these differences in positivity, in addition to the type of test performed, may also be influenced by the type of sample used (liver/intestine versus cloacal swab). Besides, they should compare their results with data from other studies.

Response: In the discussion section, Authors have explained the differences among the different samples that can be used for detection of C. psittaci infection. In the new Table 1 all these differences in positivities results are now more clear.

Line 197. The authors should add the meaning of the abbreviation “PPE”.

Response: The acronym has been explicited.

Lines 201-206. Remove this information which is already in the “Introduction section”. This information is repeated unnecessarily.

Response: information removed.

Conclusions section

Other conclusions based on the results of this study should be included in this section of the paper. For example, highlighting the samples (cloacal and nasopharyngeal swab) and tests (16S rRNA and ompA PCR) that were best for identification of the infectious agent in this outbreak.

Response: In the conclusion section more information on the samples and tests usually used for C. psittaci detection has been pointed out. Thanks to your suggestion, this section is now more complete.

Line 233. Chlamydia spp.

Response: The author changed “Chlamydia” with “Chlamydia spp.”

References section

References 1 and 15. Should the name of the journal be written in abbreviations?

Response: Abbreviations added